# Deep transfer learning for inter-chain contact predictions of transmembrane protein complexes

Peicong Lin[1], Yumeng Yan [1], Huanyu Tao[1] & Sheng-You Huang [1] ✉

Membrane proteins are encoded by approximately a quarter of human genes. Inter-chain residue-residue contact information is important for structure prediction of membrane protein complexes and valuable for understanding their molecular mechanism. Although many deep learning methods have been proposed to predict the intra-protein contacts or helix-helix interactions in membrane proteins, it is still challenging to accurately predict their inter-chain contacts due to the limited number of transmembrane proteins. Addressing the challenge, here we develop a deep transfer learning method for predicting inter-chain contacts of transmembrane protein complexes, named DeepTMP, by taking advantage of the knowledge pre-trained from a large data set of non-transmembrane proteins. DeepTMP utilizes a geometric triangle-aware module to capture the correct inter-chain interaction from the coevolution information generated by protein language models. DeepTMP is extensively evaluated on a test set of 52 self-associated transmembrane protein complexes, and compared with state-of-the-art methods including DeepHomo2.0, CDPred, GLINTER, DeepHomo, and DNCON2_Inter. It is shown that DeepTMP considerably improves the precision of inter-chain contact prediction and outperforms the existing approaches in both accuracy and robustness.

Membrane proteins (MPs) play diverse roles and essential functions in living cells including molecular transporters, ion channels, signal receptors, immune response, and enzymes. It is estimated that up to approximately a quarter of the human genome encodes membrane proteins, which constitute about half of current drug targets[1–3]. Frequently, transmembrane proteins (TMPs) assemble to form symmetric homo-oligomers to perform their specific biological functions by interacting with themselves under the drive of hydrophobic interaction and hydrogen bond networks[4,5]. For example, most of the single-spanning homo-oligomeric transmembrane complexes are stabilized by the hydrogen bonds[6], and the transmembrane region of polytopic membrane proteins would consist of hydrophobic segments with regular secondary structures[7]. However, the experimental determination of transmembrane protein complex structures is challenging, primarily due to the influence of the complicated membrane environment and the large size of these proteins. Therefore, it is highly desirable to develop computational methods for predicting the homo-oligomeric structures of TMPs and providing molecular interaction insights.

Many statistical and deep learning-based methods have been developed to predict protein structures or intra-protein contacts[8–17]. Although those methods can accurately predict the intra-protein contacts of soluble proteins, the corresponding precision for TMPs is still not satisfactory due to the different structural characteristics and physicochemical environments between soluble proteins and TMPs. Hence, some methods have been proposed to address the specific structures of TMPs[18–20]. Since most regions of TMPs are composed of α-helical structures which are mainly driven by hydrophobic interactions, several approaches use machine learning-based methods to predict the helix-helix contacts[21–24]. In addition, utilizing co-evolutionary information, Wang et al. also proposed a deep learning-based method to predict the reside-residue contacts for guiding ab initio folding of

[1]School of Physics, Huazhong University of Science and Technology, Wuhan, Hubei 430074, China. ✉e-mail: huangsy@hust.edu.cn

membrane proteins[25]. Although these methods improve the prediction of membrane protein monomer structures, it is needed to develop computational methods for addressing the oligomeric structures of TMPs and investigating their molecular interactions.

Motivated by the success of intra-protein contact predictions in monomer structure prediction[26–28], various advanced deep learning methods have been developed to predict the inter-chain contacts for protein complexes[29–40]. Our previous work, DeepHomo[29], utilizes sequence and structure features to predict inter-chain contacts with ResNet2 architectures[41]. With the progress of protein language models, DeepHomo2.0[30], GLINTER[37], and CDPred[40] apply the embedding vector and multi-head attention features from the ESM-MSA-1b model[42] to capture the interfacial interaction. However, these deep learning-based methods are only trained on the data set of mainly soluble proteins, which are different from the TMPs. Therefore, it is urgent to specially develop a deep learning model to predict the inter-chain contacts for the homo-oligomers of TMPs.

However, compared with thousands of soluble protein complexes, the number of transmembrane protein complexes is rather limited. For example, there are only <350 non-redundant homo-oligomeric transmembrane protein complexes in the PDBTM database[43], which poses a major obstacle to direct training on transmembrane proteins. Addressing the challenge, we here develop a deep transfer learning method to predict the inter-chain contacts of transmembrane proteins, named DeepTMP, which first trains an initial model on a large number of soluble proteins and then transfers it to transmembrane proteins by utilizing the features of protein language model and the geometric triangle-aware module. Compared with the common ResNet and attention mechanism, the geometric triangle module can efficiently consider the many-body effect and reduce the geometric inconsistency, which helps the pre-trained model to more effectively capture the interfacial interaction from the evolution information generated by the ESM-MSA-1b model and better predict the inter-chain contacts of TMPs.

## Results

### The overview of DeepTMP

Figure 1 shows an overview of DeepTMP, which is composed of two main stages. The first stage is to train an initial model on a large data set of homodimers which mainly consist of soluble protein complexes. The second is a transfer learning stage that fine-tunes the initial model on a small set of transmembrane protein complexes. As shown in Fig. 1a, the inputs of DeepTMP are monomer structures and the corresponding Multiple Sequence Alignment (MSA). Then, the evolutionary conservation and coevolution information are calculated from the MSA. In addition, the sequence representation and multi-head attention matrix are also generated by a protein language model. Meanwhile, the intra-protein distance map is extracted from the monomer structure. Given the preprocessed input features, the integrated features for receptor, ligand, and complex are fed into the ResNet-Inception module to extract the high-order intra-protein interactions for the receptor and ligand, and the inter-chain interaction for the complex. Then, we apply a geometric triangle-aware module to the inter-chain interaction, which considers the many-body effects by utilizing an attention mechanism on pair representations of three residues that satisfy the geometric consistency. Finally, the inter-chain contacts are predicted from the hidden representations of the complex.

### Performance of DeepTMP on transmembrane proteins

DeepTMP was extensively evaluated on a diverse test set of 52 TMP homo-oligomers from the PDBTM database with the experimental and the AlphaFold2-predicted monomer structures (Supplementary

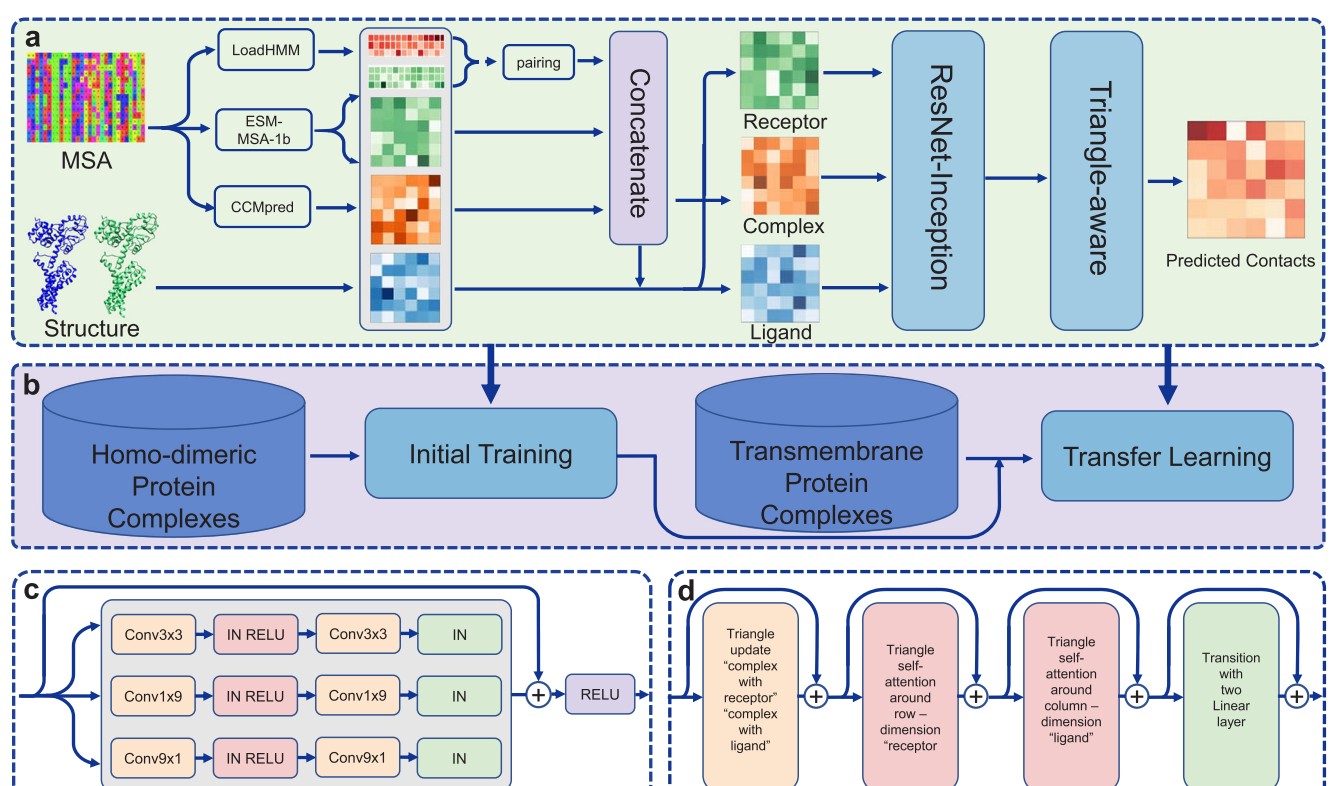

**Fig. 1 | The framework of the DeepTMP. a** The overview of the deep learning network. The input features are composed of receptor, ligand and complex features. The features of receptor are the same as ligand which includes PSSM, DCA, ESM-MSA-1b and intra-protein distance features. The features of complex exclude distance information. **b** The transfer learning protocol that includes the two stages of initial training and transfer learning. **c** The ResNet-Inception module. **d** The triangle-aware module that includes one triangle update, two triangle self-attention, and one transition.

Data 1). Table 1 lists the average precisions for the top 1, 10, 25, 50, L/10, L/5, and L predicted inter-chain contacts, where L is the length of monomer sequence. Here, the precision is defined as the percentage of the true contacts among the considered top predicted contacts.

It can be seen from the table that DeepTMP achieves a high performance for different numbers of top predicted contacts. With the input of experimental monomer structures, DeepTMP yields high precisions of 82.7%, 82.3%, 81.1%, 79.4%, 82.0%, 80.1%, and 68.4% when the top 1, 10, 25, 50, L/10, L/5, and L predicted inter-chain contacts are considered, respectively. In addition, it is often unavailable to access the experimental monomer structures in practical scenarios. Therefore, we also evaluated DeepTMP on the test set of 52 cases using the AlphaFold2-predicted monomer structures as input, whose results are also listed in Table 1. It can be seen from the table that DeepTMP still maintains a higher performance for the input of AlphaFold2-predicted monomer structures. Specifically, DeepTMP gives the precisions of 76.9%, 76.5%, 74.8%, 71.9%, 75.3%, 72.5%, and 62.3% for the top 1, 10, 25, 50, L/10, L/5, and L predicted inter-chain contacts. These results demonstrate the robustness of DeepTMP with both experimental and predicted monomer structures. Given the similar trend of DeepTMP with experimental and predicted monomer structures, we will only discuss the results with the input of experimental monomer structures in this study, unless otherwise specified.

### Transfer learning improves initial training

Transfer learning uses the knowledge from an initial model trained on large amounts of data and applies it to a related but different task with limited data. Although the membrane proteins are encoded by ~20–30% of human genes and play crucial roles in a wide range of biological processes, it is difficult to solve their structures by experiments because of their membrane environment. Hence, transfer learning can be an effective approach for the inter-chain contact prediction of transmembrane protein complexes. In addition, some studies reveal that lipid binding is not essential for the oligomerization of transmembrane proteins[44] and the residues involved in the transmembrane oligomer interface are mostly similar to those of soluble protein interfaces[45]. Therefore, the initial training model on a large set of soluble protein complexes provides a generally transferable knowledge for predicting inter-chain contacts of transmembrane protein complexes.

Table 1 lists the average precisions of the initial training model (IT_Model) that is trained on a large data set of soluble protein complexes, for several numbers of top inter-chain contacts on the test set of 52 cases (Supplementary Data 2). It can be seen from the table that DeepTMP achieves a considerable improvement of >23% compared with IT_Model for different numbers of predicted contacts. Figure 2a

shows the violin plots of the top 10 and top L precisions for DeepTMP and IT_Model. It is shown that DeepTMP obtains a median of 100% in the top 10 precisions, which outperforms that of IT_Model. In addition, DeepTMP has a lower gap between the 25th and 75th percentiles, compared with IT_Model where the 25th percentile of IT_Model is close to 0. This indicates that DeepTMP is more stable and accurate than IT_Model in predicting the inter-chain contacts of transmembrane protein complexes.

Figure 2b shows the top L precisions of DeepTMP versus IT_Model for all the cases in the test set. It can be seen from the figure that DeepTMP achieves a better performance than IT_Model for 40 of 52 cases. Figure 2c shows the top L precision gap between DeepTMP and IT_Model versus the percentage of transmembrane region for each target. Here, the sequence belonging to the transmembrane region is obtained from the PDBTM database, which is defined by the TMDET algorithm[46]. It can be seen from the figure that DeepTMP achieves a significant improvement in precision for most of the targets across the whole percentages of transmembrane regions. Such a trend can be understood because our transfer learning has been trained to improve the accuracy of DeepTMP from the IT_Model for all the targets with different transmembrane proportions. These results demonstrate the necessity of transfer learning and the robustness of DeepTMP.

Figure 2d shows the average improvement of DeepTMP over IT_Model for top L precision as a function of the number of chains for each target. It can be seen from the figure that as the number of chains increases, the precision is improved more and remains a relatively high improvement after reaching a certain level. Specifically, the improvements are 8.5%, 17.4%, 39.5%, 41.5%, 66.2%, 49.0%, 44.6%, and 62.6% for the eight chain number ranges, respectively. This phenomenon may be understood as follows. The IT_Model is only trained with homodimers, while DeepTMP is trained with different symmetries of transmembrane protein complexes consisting of different numbers of chains. On one hand, IT_Model does not learn the high-order symmetric inter-chain interactions during the training process. On the other hand, the formation of homo-oligomers with more chains would generate a more negative entropy[47]. To maintain the stability of the assembly, it is expected that its sequence should be more evolutionarily conserved than homodimers.

### Transfer learning outperforms direct training

To investigate the importance of the dataset of soluble protein complexes, we directly trained a model on the training set of transmembrane protein complexes with the same network and hyperparameters as DeepTMP, named the direct training model (DT_Model). Table 1 lists the average precisions of DT_Model for several numbers of top predicted contacts on the test set of 52 targets

### Table 1 | Comparison of DeepTMP with other state-of-the-art methods

| Method | Top 1 | Top 10 | Top 25 | Top 50 | Top L/10 | Top L/5 | Top L |
|---|---|---|---|---|---|---|---|
| **DeepTMP** | **82.7 (76.9)** | **82.3 (76.5)** | **81.1 (74.8)** | **79.4 (71.9)** | **82.0 (75.3)** | **80.1 (72.5)** | **68.4 (62.3)** |
| CDPred | 55.8 (46.2) | 48.5 (49.6) | 48.4 (49.2) | 45.9 (46.5) | 48.6 (49.0) | 45.6 (44.8) | 33.8 (35.3) |
| DeepHomo2.0 | 50.0 (46.2) | 48.7 (46.7) | 47.2 (44.9) | 43.3 (41.1) | 44.0 (42.0) | 41.1 (38.7) | 31.8 (27.2) |
| GLINTER | 44.4 (40.0) | 38.0 (36.4) | 38.3 (36.6) | 34.3 (32.8) | 39.9 (36.0) | 36.4 (34.7) | 27.7 (27.4) |
| DeepHomo | 28.8 (23.1) | 27.7 (23.3) | 24.8 (20.5) | 22.0 (19.2) | 23.2 (18.7) | 21.4 (18.4) | 15.7 (12.1) |
| DNCON2_Inter | 13.7 (13.7) | 13.3 (13.1) | 12.0 (13.0) | 11.7 (13.0) | 12.5 (12.8) | 11.4 (12.7) | 7.9 (8.3) |
| DeepHomo2_TMP | 63.5 (61.5) | 65.0 (63.3) | 63.0 (59.7) | 60.9 (56.2) | 60.6 (57.1) | 58.5 (55.9) | 45.9 (40.9) |
| IT_Model | 59.6 (57.7) | 56.3 (55.8) | 54.8 (53.8) | 52.3 (51.0) | 53.9 (52.9) | 52.4 (50.9) | 42.0 (39.3) |
| DT_Model | 51.9 (55.8) | 53.5 (53.3) | 53.4 (53.2) | 50.2 (49.0) | 50.4 (51.5) | 49.9 (47.9) | 38.5 (37.4) |

The precisions (%) of DeepTMP, CDPred, DeepHomo2.0, GLINTER, DeepHomo, and DNCON2_Inter are based on the test set of 52 transmembrane protein complexes when the experimental monomer structures (predicted monomer structures by AlphaFold2) are used as input. For reference, the table also lists the results of the transfer learning model using the network architecture of DeepHomo2.0 (DeepHomo2_TMP), the initial training model (IT_Model) on the large data set of soluble protein complexes, and the direct training model (DT_Model) on the small data set of transmembrane protein complexes. The numbers in bold fonts indicate the best performances for the corresponding categories.

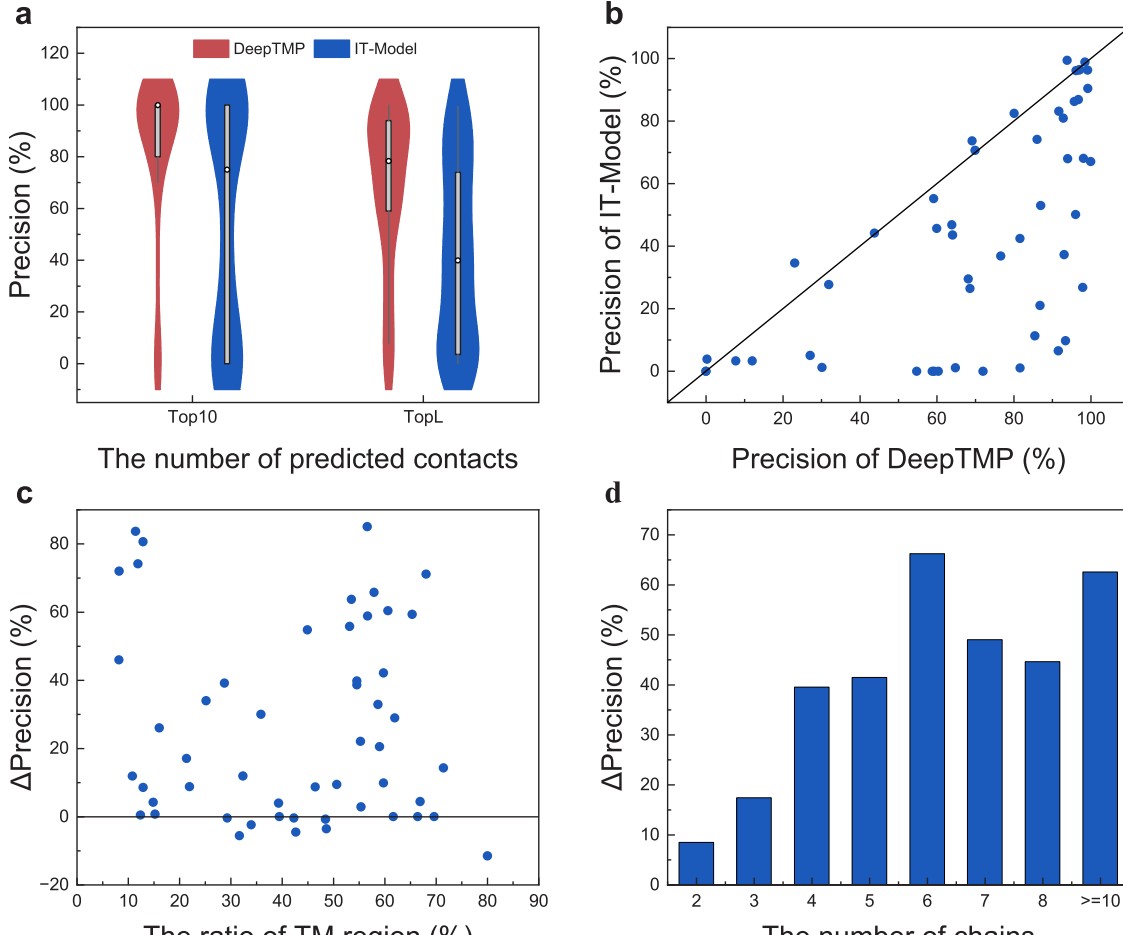

**Fig. 2 | Comparison of DeepTMP and initial training model (IT_Model). a** The violin plots of the top 10 and L precisions for DeepTMP and the initial training model (IT_ Model) on *n* = 52 cases in the TMP test set. The box plot inside the violin displays the 25th and 75th percentiles of the data, and the dot indicates the median. The whiskers extend to the highest and lowest values within 1.5 times the interquartile range of the data. **b** The top L precisions of DeepTMP versus those of IT_Model for all 52 test cases. **c** The gap of top L precision between DeepTMP and IT_Model versus the percentage of transmembrane region on the test set of 52 targets. **d** The gap of top L precision between DeepTMP and IT_Model as a function of the number of chains in each target.

(Supplementary Data 2). The corresponding comparison between DeepTMP and DT_Model for top 10 and L precisions is shown in Fig. 3a. DeepTMP exhibits an improvement of ~30% across different numbers of predicted contacts in comparison with DT_Model. The precisions of DeepTMP maintain a higher level. This indicates that it is difficult for DT_Model to directly learn the inter-chain interaction of transmembrane protein complexes from a small training set.

Figure 3b shows the top L precisions of DeepTMP versus those of DT_Model. It can be seen from the figure that DeepTMP gives a better performance than DT_Model for 44 of 52 test cases. Figure 3c shows the top L precision improvement of DeepTMP over DT_Model versus the proportion of transmembrane regions. It can be seen from the figure that DeepTMP achieves a considerable improvement over DT_Model for the majority of the cases for different percentages of transmembrane regions. The improvement is much more prominent when the percentage of transmembrane region becomes smaller for a target. This can be understood because the targets with smaller transmembrane proportions will be more like soluble proteins which are not considered by DT_Model. These results suggest that the improvement may be due to the abundant similar interactions between soluble and transmembrane protein complexes, such as the interfacial hydrophobic interactions. It is apparent that DeepTMP can effectively retain the similar interaction knowledge between soluble and transmembrane protein complexes through the transfer learning

protocol, but DT_Model fails to do so because it is only trained on the small set of transmembrane proteins.

To verify whether IT_Model and DT_Model learn the characteristics of soluble and transmembrane proteins, respectively, we compare the precisions of IT_Model and DT_Model for the top L predicted contacts, as shown in Fig. 3d. Overall, the performances of the two methods are comparable, albeit with a discernible advantage for one method over the other depending on the performance of IT_Model. Specifically, DT_Model has a lower precision than IT_Model for the targets where IT_Model has a relatively high precision (above ~60%), while DT_Model obtains a better performance than IT_Model on those targets where IT_Model has a relatively low precision (below ~40%). Such a trend can be understood as follows. For the targets where the IT_Model has a high performance, their inter-chain interactions would be more like those of soluble proteins. As such, DT_Model would perform less satisfactorily on these targets because it is only trained on transmembrane proteins. In contrast, for the targets where the IT_Model has a low performance, their inter-chain interactions would be more impacted by the membrane environment. Therefore, it is expected that DT_Model performs better than IT_Model on these transmembrane protein complexes. The different advantages of IT_Model and DT_Model ensure the robustness of DeepTMP in improving the inter-chain contact predictions of transmembrane protein complexes through a transfer learning strategy.

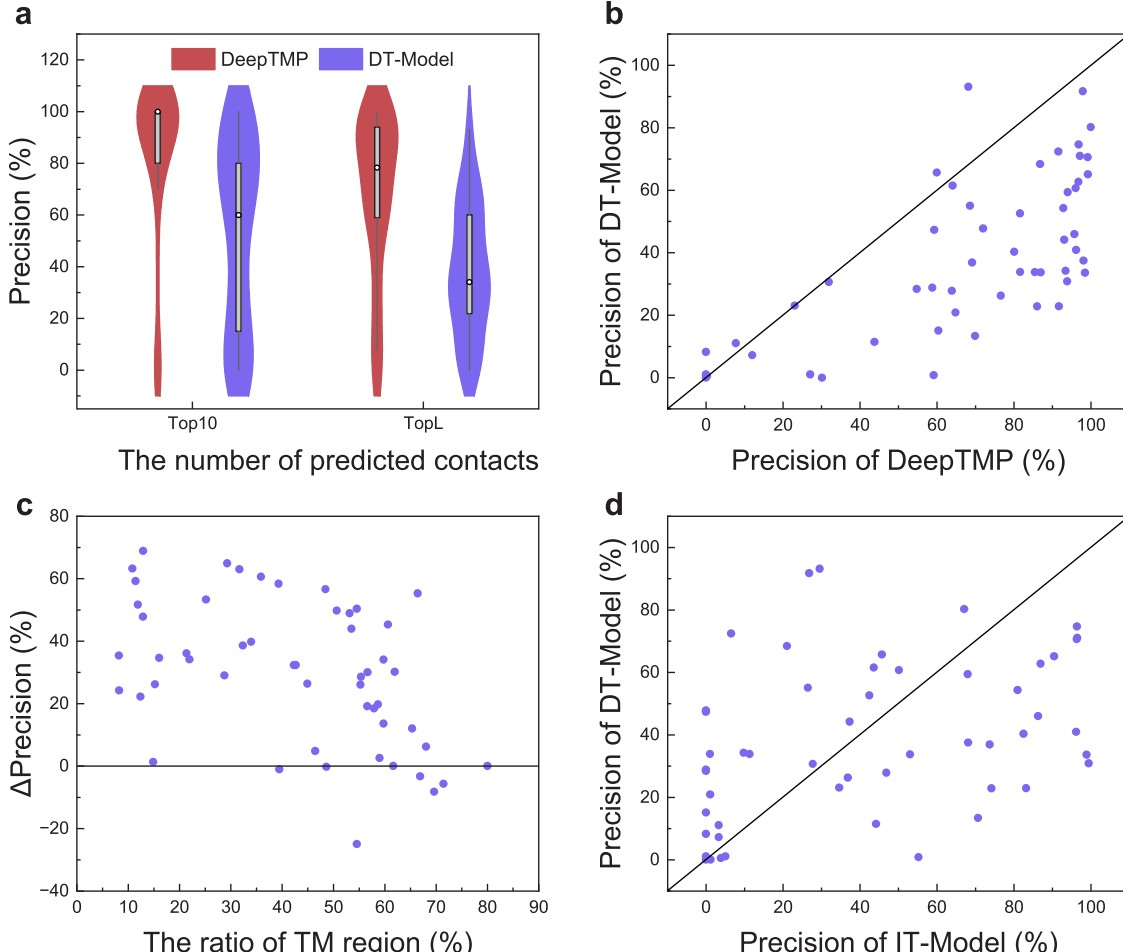

**Fig. 3 | Comparison of DeepTMP and direct training model (DT_Model). a** The violin plots of the top 10 and L precisions for DeepTMP and the direct training model (DT_ Model) on $n = 52$ cases in the TMP test set. The box plot inside the violin displays the 25th and 75th percentiles of the data, and the dot indicates the median. The whiskers extend to the highest and lowest values within 1.5 times the interquartile range of the data. **b** The top L precisions of DeepTMP versus those of DT_Model for all 52 test cases. **c** The gap of top L precision between DeepTMP and DT_Model versus the percentage of transmembrane region on the test set of 52 targets. **d** The top L precisions of IT_Model versus those of DT_Model for all 52 transmembrane protein complexes in the test set.

## Comparison with other methods

We compared DeepTMP with other inter-chain contact predictors including CDPred, DeepHomo2.0, GLINTER, DeepHomo, and DNCON2_Inter on the test set of 52 targets with experimental and AlphaFold2-predicted monomer structures, respectively. Specifically, we downloaded the programs of the methods other than DeepHomo2.0 and DeepHomo from their official web sites, and then ran the programs locally using their default parameters. It is noted that GLINTER only gave the results for 45 targets for the TMP test set because it can only handle the targets with a sequence length of <500. Table 1 lists the average precisions of the five methods for the top 1, 10, 25, 50, L/10, L/5, and L predicted inter-chain contacts on the test set of 52 targets (Supplementary Data 1).

It can be seen from the table that DeepTMP significantly outperforms the other five methods for different numbers of predicted contacts. Specifically, DeepTMP achieves a precision of 82.3% for the top 10 predicted inter-chain contacts with the input of experimental monomer structures, which is much >48.7% for DeepHomo2.0, 48.5% for CDPred, 38.0% for GLINTER, 27.7% for DeepHomo, and 13.3% for DNCON2_Inter. Similar trend can also be observed for the input of AlphaFold2-predicted monomer structures. Specifically, DeepTMP gives the precisions of 76.5% and 62.3% for top 10 and L predicted contacts, which are considerably >49.6% and 35.3% for CDPred, 46.7% and 27.2% for DeepHomo2.0, 36.4% and 27.4% for GLINTER, 23.3% and

12.1% for DeepHomo, and 13.1% and 8.3% for DNCON2_Inter. Given the difference in structure characteristics between soluble and transmembrane protein complexes, it is expected that DeepTMP performs better than DeepHomo2.0, CDPred, DeepHomo, and DNCON2_Inter which are mainly trained on soluble protein complexes. This can also be understood because DeepTMP extracts the high-order representations of transmembrane protein complexes through transfer learning and learns the interaction knowledge of both two soluble and transmembrane protein systems.

Figure 4a shows a comparison of the top L precisions of DeepTMP, CDPred, DeepHomo2.0, GLINTER, DeepHomo, and DNCON2_Inter in terms of median, 25th, and 75th percentiles. It can be seen from the figure that DeepTMP achieves the highest median top L precision among the six methods. In addition, it is also noted that the 25th percentile of DeepTMP is higher than the 75th percentile of the other methods. This suggests that DeepTMP is more accurate and robust than the other methods in inter-chain contact predictions on transmembrane protein complexes. Figure 4b shows the top L precisions of DeepTMP versus those of CDPred, DeepHomo2.0, GLINTER, DeepHomo, and DNCON2_Inter for the 52 cases in the test set. It can be seen from the figure that DeepTMP achieves a significantly better performance than the other five methods for most of the test cases. Specifically, DeepTMP performs better than CDPred for 47 of 52 cases, better than DeepHomo2.0 for 46 of 52 cases, better than GLINTER for

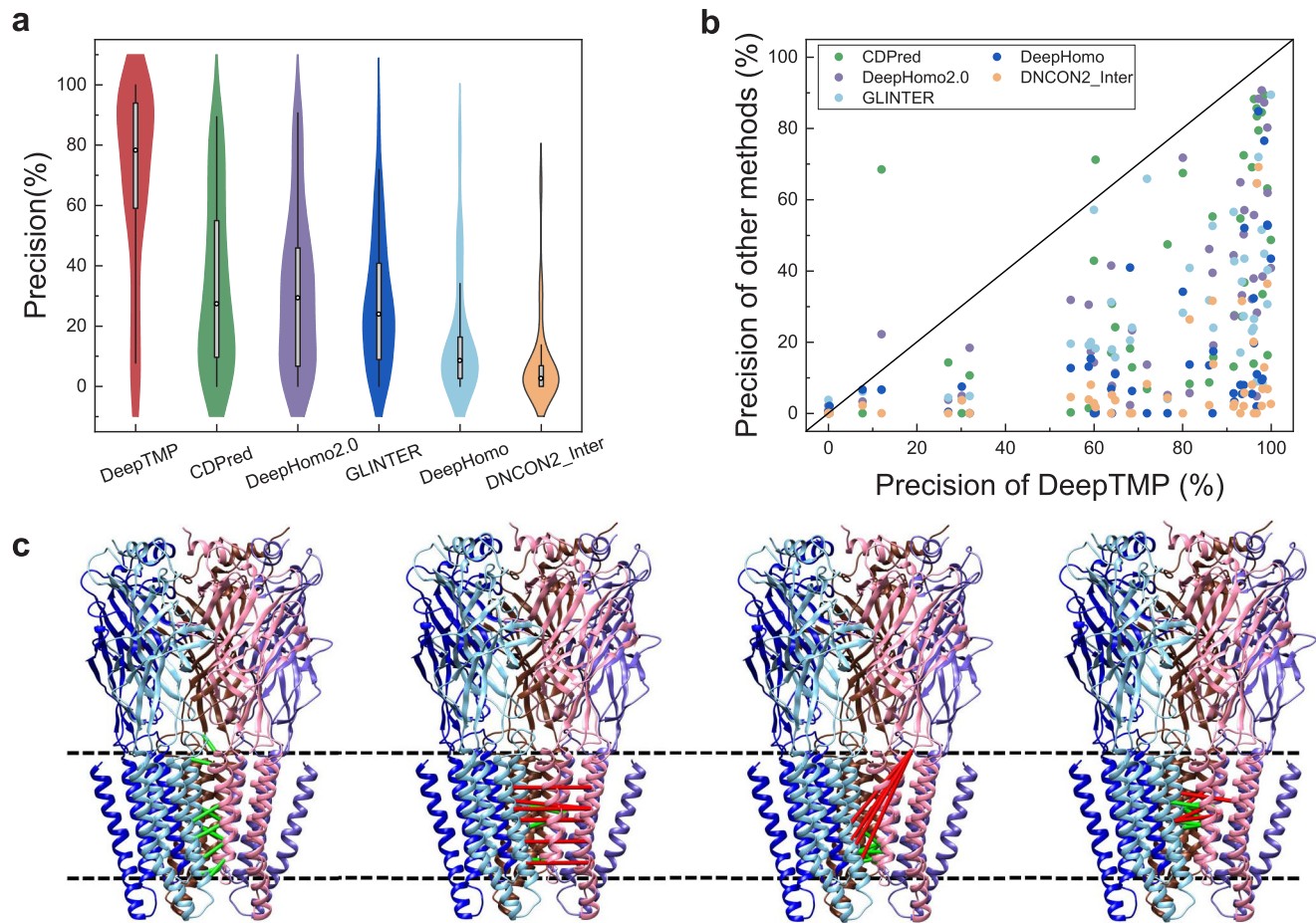

**Fig. 4 | Comparison of DeepTMP and the stat-of-the-art methods. a** The violin plots of top L precisions by DeepTMP, CDPred, DeepHomo2.0, GLINTER, DeepHomo and DNCON2_Inter on *n* = 52 cases in the TMP test set. The box plot inside the violin displays the 25th and 75th percentiles of the data, and the dot indicates the median. The whiskers extend to the highest and lowest values within 1.5 times the interquartile range of the data. **b** The top L precisions of DeepTMP versus those

of CDPred, DeepHomo2.0, GLINTER, DeepHomo, and DNCON2_Inter on the test set. **c** The 3D structures of 5TIN with predicted inter-chain contacts by DeepTMP, CDPred, DeepHomo2.0, and GLINTER. The correct (incorrect) predicted contacts are colored in green (red). The parts between two dashed lines are the transmembrane regions.

41 of 45 cases, better than DeepHomo for 47 of 52 cases, and better than DNCON2_Inter for 46 of 52 cases. These results indicate that DeepTMP can more accurately predict the inter-chain contacts of transmembrane protein complexes.

Figure 4c shows an example of transmembrane protein complex (PDB ID: 5TIN) with the predicted inter-chain contacts by four methods. Although the helix-helix interactions in the transmembrane region belong to the hydrophobic interactions which may be similar to those at the interface of soluble protein complexes, CDPred, DeepHomo2.0, and GLINTER still cannot well predict the inter-chain contacts. In contrast, DeepTMP obtains high precisions of 100% and 87% for top 10 and top L predictions, respectively. This demonstrates that DeepTMP can correctly capture the hydrophobic interaction knowledge at the interface of transmembrane protein complexes which is affected by the lipid environment.

To further examine the robustness of DeepTMP, we also evaluated the performance of DeepTMP on the 296 soluble dimeric proteins from the test set of DeepHomo2.0, and compared it with CDPred, DeepHomo2.0, GLINTER and DeepHomo. Supplementary Table 1 lists the corresponding precisions for the top 1, 10, 25, 50, L/10, L/5, and L predicted inter-chain contacts on the test set of 296 soluble dimers (Supplementary Data 3). It can be seen from the table that DeepTMP still maintains a good performance and obtains the precisions of 71.2% and 65.1% for top 10 and L predicted contacts on these soluble

proteins, which are comparable to the precisions for CDPred and significantly higher than those for the other methods. This suggests that DeepTMP can well retain the learned inter-chain contact knowledge of soluble protein complexes from the initial training stage during transfer learning to transmembrane protein complexes.

### Analysis of feature importance

The number of interfacial contacts in a protein complex is an important feature of its inter-chain interaction. Here, the contact density is used to characterize the interface. Specifically, the number of inter-chain contacts is first obtained from the maximum interface in a target. Then, the contact density is calculated as follows,

$$\text{ContactDensity} = \frac{\text{sum(contacts)}}{2L} \tag{1}$$

where *L* is the sequence length of monomer structure for transmembrane protein complexes. Figure 5a shows the top L precisions of the six methods versus the contact density in different ranges on the TMP test set. It can be seen from the figure that DeepTMP achieves the best performance for different contact densities among all the tested methods. In general, there is an uptrend of precision when the contact density increases, as expected. Since most transmembrane protein complexes are composed of a large number of amino acids, it is

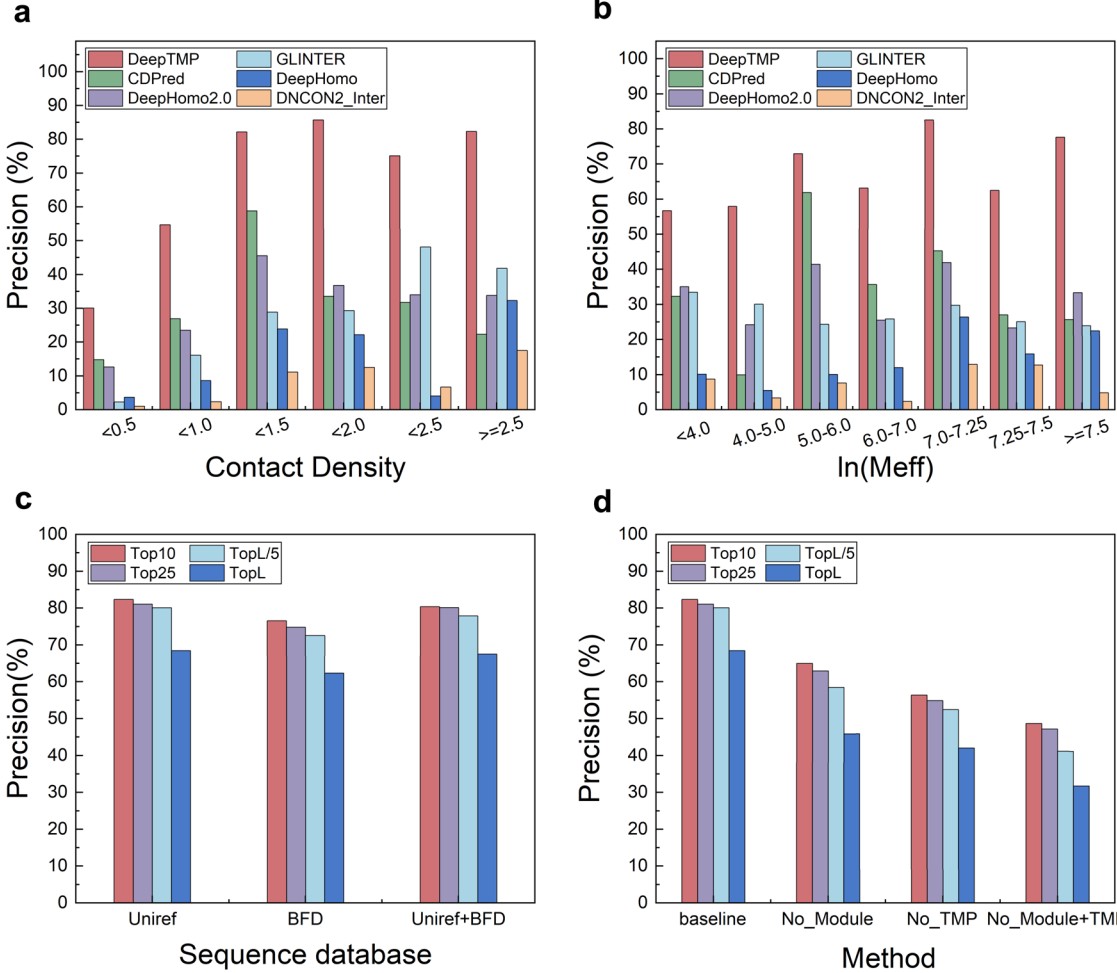

**Fig. 5 | Analysis of feature importance. a** The top L precisions of DeepTMP, CDPred, DeepHomo2.0, GLINTER, DeepHomo, and DNCON2_Inter in different ranges of contact density. **b** The top L precisions of DeepTMP, CDPred, DeepHomo2.0, GLINTER, DeepHomo, and DNCON2_Inter in different ranges of ln($M_{eff}$). **c** Comparison of the precisions for DeepTMP using UniRef, BFD, and UniRef+BFD sequence databases, respectively. **d** The precisions of the baseline model (i.e. DeepTMP) versus three ablation models including No_Module (i.e. DeepHomo2_TMP), No_TMP (i.e. IT_Model), and No_Module+TMP (i.e. DeepHomo2.0).

difficult to capture the small portion of interfacial interactions. Nevertheless, even at a low contact density, DeepTMP obtained a top L precision of 30%, which is significantly higher than the other methods (Supplementary Data 4a).

We further investigated the influence of the depth of MSA on the performances of DeepTMP and other five methods. The effective sequence number $M_{eff}$ is used to quantify the depth of MSA. Specifically, we used a sequence identity of 70% to calculate the $M_{eff}$ on the TMP test set and set seven reasonable intervals. Figure 5b shows the corresponding top L precisions of the six methods in different ranges of MSA depth. It can be seen from the figure that DeepTMP also achieves the best performance among the six methods. Even at a lower value of ln($M_{eff}$) < 4.0, DeepTMP is capable of achieving a top L precision of over 55%, which is comparable to the best performance of the other methods, surpassing most of the other methods by a significant margin.

Interestingly, the targets with the ln($M_{eff}$) from 6.0 to 7.0 have a relatively lower precision for all methods (Fig. 5b). Further examination reveals that two alpha targets (PDB IDs: 4JKV and 5GUF) in this MSA range have low contact density of 0.343 and 0.324 (Supplementary Data 4a), which makes it hard to correctly predict the inter-chain contacts. Similarly, another alpha target 4OR2 also has a contact density of 0.325, which leads to a relatively low average precision for the ln($M_{eff}$) from 7.0 to 7.25. If we excluded these three targets from

the test set, the precisions will always increase with the increase of ln($M_{eff}$). Another reason for such an abnormal phenomenon is that we used hhfilter to screen the sequences of the original MSA for extracting the ESM-MSA-1b features, which often leads to a drop in the number of effective sequences for ESM-MSA-1b. Therefore, we have instead adopted the number of effective sequences for ESM-MSA-1b to measure the depth of MSA. With this correction, the precision indeed increases with the number of effective homologous sequences (Supplementary Data 4a).

The searched sequence database is one critical factor impacting the MSA. Therefore, we have also evaluated DeepTMP using the Big Fantastic Database (BFD) instead of the original UniRef_2020_03 database. Figure 5c shows a comparison of DeepTMP with UniRef, BFD, and BFD+UniRef databases in terms of precisions on the test set of 52 targets (Supplementary Data 4b and Supplementary Data 4c). Interestingly, DeepTMP/BFD yields a slightly lower performance than the baseline DeepTMP/UniRef model, although the DeepTMP/UniRef+BFD model recovers the lost performance. Specifically, DeepTMP/BFD gives the precisions of 77.5% and 65.6% for top 10 and L contacts, compared with 82.3% and 68.4% for DeepTMP/UniRef, and 80.4% and 67.5% for DeepTMP/UniRef+BFD (Supplementary Data 4b). The reason for such difference would be attributed to the different preparation methods between UniRef and BFD databases, which will result in

different MSAs for the same query. It is also revealed that DeepTMP/ BFD is quite complementary to DeepTMP/UniRef. Specifically, out of the 52 targets, DeepTMP/BFD performs better for 22 cases, DeepTMP/ UniRef performs better for 24 cases, and both models perform comparably for 6 cases (Supplementary Data 4b).

Compared with other methods like DeepHomo2.0, DeepTMP has two additional types of important features. One is the use of Resnet-Inception and geometric triangle-aware modules in the network architecture, and the other is the inclusion of transmembrane protein complexes in the training set. To investigate their roles in DeepTMP, we have retrained a model using the same network as DeepHomo2.0 and the same training procedure as DeepTMP, named DeepHomo2_TMP. Figure 5d shows a comparison of the baseline model (i.e. DeepTMP) and three ablation models including No_Module (i.e. DeepHomo2_TMP), No_TMP (i.e. IT_Model), and No_Module+TMP (i.e. DeepHomo2.0) on the test set of 52 targets (Supplementary Data 5). It can be seen from the figure that all the ablation models give a lower performance than the baseline model, as expected. However, compared with the baseline, the No_TMP model has a larger performance drop than the No_Module model. For example, DeepTMP (i.e. baseline) gives a precision of 82.0% for top L/10 contacts, compared with 53.9% for IT_Model (No_TMP), 60.6% for DeepHomo2_TMP (i.e. No_Module), and 44.0% for DeepHomo2.0 (i.e. No_Module+TMP). These results suggest that the transmembrane protein complexes contribute more to the performance of DeepTMP than the Resnet-Inception and geometric triangle-aware modules.

Table 1 also shows that even the IT_Model yields a better performance than the other methods including CDPred, DeepHomo2.0, GLINTER, DeepHomo, and DNCON2_Inter, although these models are all trained on soluble protein complexes. Such phenomenon can be understood as follows. Compared with the other methods, the IT_Model implements the Resnet-Inception and geometric triangle-aware modules in the network. On one hand, the Resnet-Inception module can capture long-range interaction by increasing the effective receptive field. On the other hand, the geometric triangle-aware module is also able to consider many-body effects by utilizing an attention mechanism on pair representations of three residues that satisfy the geometric consistency. As such, the IT_Model has a better ability to capture the inter-chain contacts than the other methods.

## Impact on protein topology

There are 75/12/13, 25/7/3, and 38/3/11 cases for alpha/beta/alpha+beta topologies in the training, validation, and test sets, respectively. Although alpha-helix and beta-barrel transmembrane proteins span and interact with lipid bilayer through hydrophobic interactions, their topologies, function, and interaction mechanism are different. Most transmembrane protein complexes belong to the alpha-helix category, where the alpha-helical domains are mostly hydrophobic and non-polar. Therefore, the intra-protein and inter-chain interactions are similar and hydrophobic for the alpha-helix transmembrane proteins. However, the outside surface of beta-barrel is non-polar, and the inside channel is hydrophilic, which is different from those of alpha-helix transmembrane proteins. Therefore, it is valuable to investigate the impact of different topologies of transmembrane protein complexes on inter-chain contact predictions.

Figure 6a shows the top 10, 25, L/5, and L precisions of DeepTMP for alpha, beta, and alpha+beta targets in the test set of 52 transmembrane protein complexes. It can be seen from the figure that DeepTMP obtains a better performance on the targets with beta and alpha+beta topologies than on those with alpha topology. Specifically, DeepTMP obtains the precisions of 76.6% and 62.1% for top 10 and L precisions on the targets with alpha topology, compared with 100% and 87.9% for beta-topology targets, and 97.3% and 85.0% for alpha +beta-topology targets. To investigate how the topology of the training set impacts the performances of different topologies, we retrained

a model using the alpha-transmembrane proteins only, named DeepTMPα, and then evaluated DeepTMPα on alpha, beta, and alpha +beta targets (Supplementary Data 6). Interestingly, DeepTMPα does not perform the best on those alpha targets (Fig. 6b). Similar trend in the performances of different topologies for DeepTMP can be observed for DeepTMPα. Namely, DeepTMPα achieves the best performance on beta targets, followed by alpha+beta and alpha targets (Fig. 6a, b). Specifically, DeepTMPα obtains the precisions of 58.2%, 74.5%, and 76.9% for top L predicted contacts on the alpha, beta, and alpha+beta targets, respectively (Supplementary Table 2).

This topology-dependent phenomenon may be understood as follows. Generally, beta-barrel transmembrane proteins play a crucial role in transporting cargo and signaling across biological membranes. The efficient execution of these functions necessitates the formation of multi-chain structures with a barrel-like architecture, which may lead to a large interacting interface. Thus, we calculated the average contact density of the two kinds of topologies. It is found that the targets with alpha and beta topologies have 1.33 and 2.33 contact densities, respectively. Therefore, it is expected to predict inter-chain contacts on the beta-topology targets more easily. In addition, beta-barrels often exist in the outer membranes of Gram-negative bacteria and chloroplast and mitochondria[48]. If the MSA of beta-barrel transmembrane protein complexes displays a greater degree of conservation, it is expected to have an enhanced prediction of inter-chain contacts on such kind of topology. Hence, we simply employ an entropy formula as a straightforward measurement of the MSA conservation. Specifically, the average entropy for each target is calculated as follows,

$$ H = -\frac{1}{N}\sum_{i}^{N}\sum_{j}^{n_{aa}} p_i^j \log\left(p_i^j\right) \qquad (2) $$

where $N$ is the number of amino acids in the sequence of a target, $n_{aa}$ is the 20 standard amino acids, and $p_i^j$ represents the observed frequency of amino acid type $j$ at the $i$-th position.

Figure 6c shows the distribution of average entropy on the three types of topologies for the test set of 52 targets (Supplementary Data 7). It can be seen from the figure that the targets with beta topology have a higher sequence conservation. Therefore, it is expected to predict the inter-chain contact with a better performance on beta topology. Similarly, the targets with alpha+beta topology mostly form the ion channel and beta barrel channel according to the secondary structures of their transmembrane regions. Considering the contact density and sequence conservation, the target of alpha+beta topology is expected to obtain a better performance than that of alpha topology. In addition, we also calculated the precisions of the other methods on three types of topologies. Specifically, DeepTMP achieves the top L precisions of 62.1%, 87.9%, and 85.0% on alpha, beta and alpha +beta topologies, respectively, which significantly outperform the precisions of 36.6%, 22.5%, and 27.2% for CDPred, 30.7%, 28.4%, and 36.3% for DeepHomo2.0, 23.2%, 37.1%, and 39.4% for GLINTER, and 14.2%, 5.0% and 23.7% for DeepHomo (Supplementary Data 1a). This suggests that DeepTMP learned the different characteristics for different topologies of transmembrane protein complexes, and can robustly predict their inter-chain interactions.

Transmembrane proteins consist of three different regions, including extracellular (Extra), transmembrane (TM), and cytoplasmic (Cyto) segments. Among these three categories, the transmembrane segment is embedded in the lipid bilayer, while extracellular and cytoplasmic segments are located in water and belong to soluble parts. Given the different physiochemical environments of extracellular, transmembrane, and cytoplasmic regions, we have calculated the precisions of DeepTMP for the three topologies of transmembrane proteins on the 52 targets (Supplementary Data 8), where the positions of TM regions are defined according to the PDBTM database[43]. The corresponding results are shown in

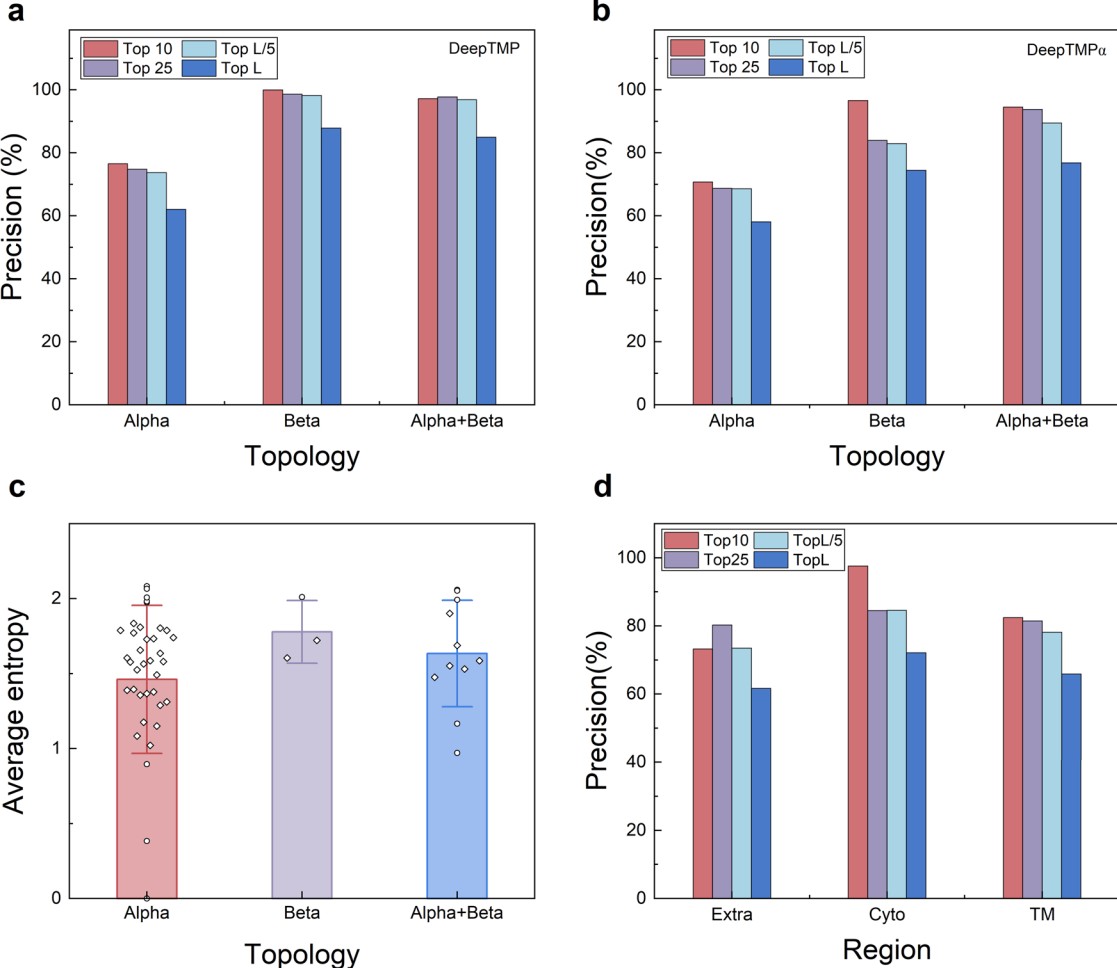

**Fig. 6 | Impact of protein topology. a** The top 10, 25, L/5, and L precisions of DeepTMP for the targets with alpha, beta, and alpha+beta topologies. **b** The top 10, 25, L/5, and L precisions of DeepTMPα that is trained on alpha transmembrane proteins only, for the targets with alpha, beta, and alpha+beta topologies. **c** The average entropies of the targets for three types of topologies including $n = 38$ alpha, 3 beta, and 11 alpha+beta cases. The vertical bars represent the mean, and the lines are error bars (standard deviations). The outliers are plotted as circle. **d** The top 10, 25, L/5, and L precisions of DeepTMP for extracellular (Extra), transmembrane (TM), and cytoplasmic (Cyto) regions of transmembrane protein.

Fig. 6d and listed in Supplementary Table 3. It can be seen from the figure that DeepTMP achieves the overall best performance with a top L precision of 72.1% for the cytoplasmic region, followed by 66.0% for the transmembrane and 61.7% for the extracellular regions. These results suggest that the inter-chain contacts for both transmembrane and non-transmembrane regions would be evolutionarily conserved in the formation of transmembrane protein complexes and thus can be well predicted by DeepTMP.

**Prediction of oligomeric state**

DeepTMP is developed to predict the inter-chain contacts of transmembrane protein complexes. Another critical challenge in the field is to predict the oligomeric state of a protein. To examine whether DeepTMP can be used for such purpose, we have investigated the maximum predicted contact probabilities of 50 monomers and 52 oligomers from our test set. Here, the 50 monomeric transmembrane proteins are obtained from the PDBTM database[43] according to the following criteria: (i) The protein only has one chain; (ii) The length of sequence is <800. Then, we used the MMseqs2 with an E-value of 0.1 to remove the overlapped monomeric transmembrane proteins with our TMP test set. Finally, we used a sequence identity cutoff of 40% to cluster the remained monomeric transmembrane proteins and randomly selected 50 monomer targets.

Figure 7a shows the box plots of the average maximum contact probabilities for 50 monomeric and 52 oligomeric transmembrane proteins (Supplementary Data 9). It can be seen from the Fig. 7a that compared with the monomers, the oligomers tend to have a higher maximum contact probability. On average, the oligomers have an average maximum contact probability of 0.86, which is significantly >0.64 for the 50 monomers. This suggests that DeepTMP can to some extent distinguish between monomers and oligomers (dimer, trimer, etc.) according to the maximum contact probability. In addition, among the oligomers, although the complexes with three or more subunits have comparable maximum contact probabilities, they have a higher maximum contact probability than the dimers (Fig. 7b). This means that DeepTMP can to some extent distinguish between dimers and the complexes with three or more subunits, but may not distinguish between the oligomeric states of those complexes with three or more subunits.

Figure 7c shows the success rates for the prediction of monomeric and oligomeric states by DeepTMP as a function of the contact probability threshold. Here, a successful prediction for a monomer is defined if its maximum contact probability is lower than the contact probability threshold, while a successful prediction for an oligomer means that its maximum contact probability is higher than the contact probability threshold. It can be seen from the figure that with the increase of

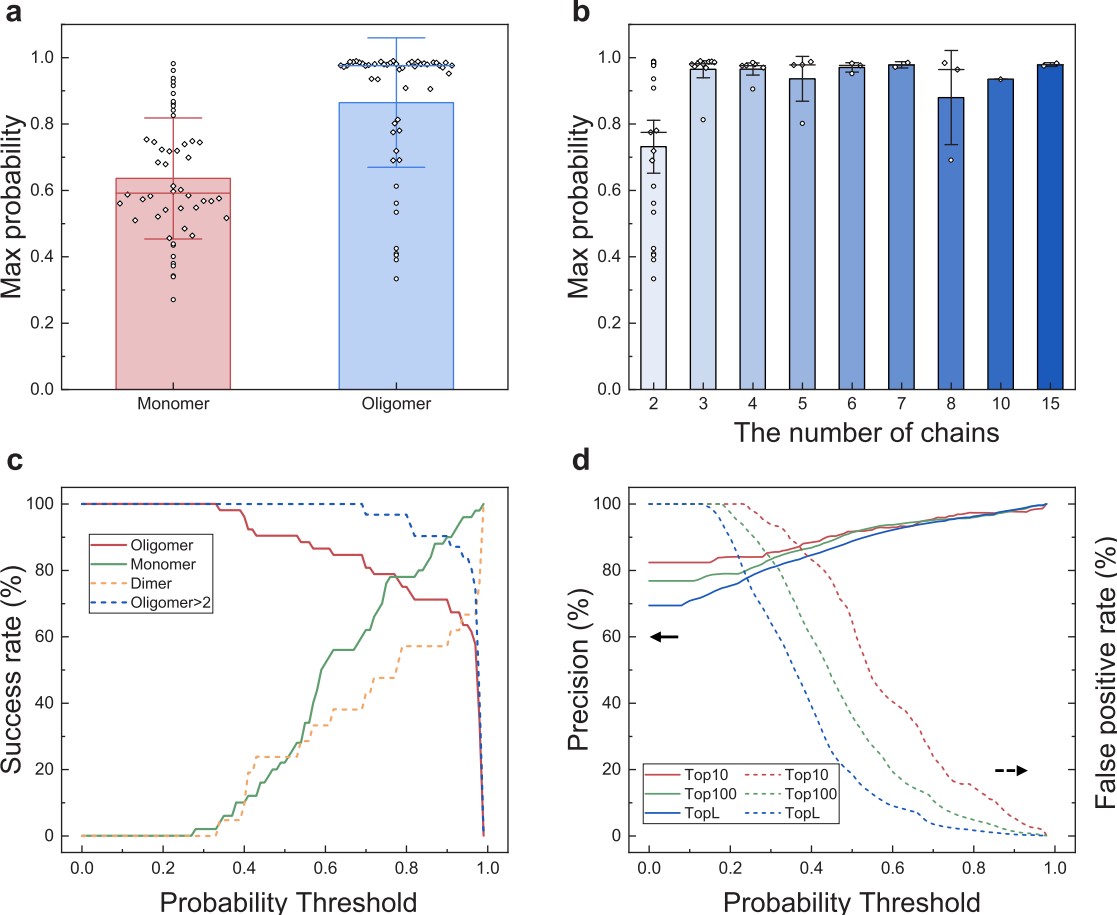

**Fig. 7 | Prediction of the oligomeric state. a** The average maximum predicted contact probabilities for $n = 50$ monomers from the PDBTM database and $n = 52$ oligomers in the TMP test set. The vertical bars represent the mean, and the lines are error bars (standard deviations). The outliers are plotted as circle. **b** The average maximum predicted contact probabilities for different oligomeric states for $n = 52$ transmembrane protein complexes in the test set. The vertical bars represent the mean, and the lines are error bars (1.5 times standard error). **c** The success rates in predicting the oligomeric states of monomer versus oligomer, and dimer versus oligomer>2 as a function of contact probability threshold. **d** The precisions for oligomers (solid lines) and the false positive rates for monomers (dashed lines) as a function of contact probability threshold.

contact probability threshold, the success rate for monomers becomes higher, while the success rate for oligomers becomes lower. When they meet at the contact probability threshold of around 0.77, both monomers and oligomers achieve a relatively high success rate. Therefore, the contact probability threshold of 0.77 may be used as a criterion to distinguish between monomers and oligomers. Similarly, the contact probability threshold of 0.97 can be used as a criterion to distinguish between dimers and oligomers with three or more subunits. Figure 7d shows the precisions of predicted contacts for oligomers and the false positive rates of predicted contacts for monomers by DeepTMP as a function of the contact probability on the 50 monomers and 52 oligomers. It can be seen from the figure that DeepTMP yields a high precision of 95.8% for oligomers and a very low false positive rate of 2.01% for monomers within top L predicted contacts at the contact probability threshold of 0.77, suggesting the potential of DeepTMP in distinguishing between monomers and oligomers.

**Case study**

Figure 8a–c shows the predicted inter-chain contact maps by DeepTMP on three selected transmembrane protein targets, 6UWM, 1UUN, and 5SVK. For comparison, the corresponding contact maps for AlphaFold-Multimer (AFM) and native complexes are also given in the figure. Here, the contact probability map for AFM is converted from the predicted inter-chain distances in the AFM model with the best "iptm+ptm" score.

It can be seen from the figure that DeepTMP obtains a better prediction with the precisions of 62.7%, 68.5%, and 93.1% for the top L contacts on these three targets, respectively, compared with 36.6%, 52.2%, and 89.0% for AFM. In addition, compared with the ground-truth contact map, DeepTMP also shows a good complementarity to AFM in terms of predicted inter-chain contacts. In other words, DeepTMP is able to predict correct contacts that are failed by AFM. For example, within the top 100 predicted contacts, DeepTMP alone gives the correct prediction for 42 and 49 contacts, AFM alone gives the correct prediction for 24 and 12 contacts, and both models give the correct prediction for 34 and 26 contacts on targets 1UUN and 6UWM, respectively.

With the predicted inter-chain contacts, we also constructed the complex structure of transmembrane proteins by using our protein-protein docking program HSYMDOCK[49]. Specifically, the putative binding models are first sampled and then evaluated by our iterative knowledge-based scoring function ITScorePP[50]. The predicted inter-chain contacts by DeepTMP are integrated into the scoring process as follows[30]:

$$E_{cont} = \begin{cases} E_0, & \text{if } r_{ij} \le 8.0\,\text{Å} \\ E_0 \times \left(1 - \frac{r_{ij} - 8.0}{4.0}\right), & \text{if } 8.0\,\text{Å} < r_{ij} \le 12.0\,\text{Å} \\ 0.0, & \text{if } r_{ij} > 12.0\,\text{Å} \end{cases} \quad (3)$$

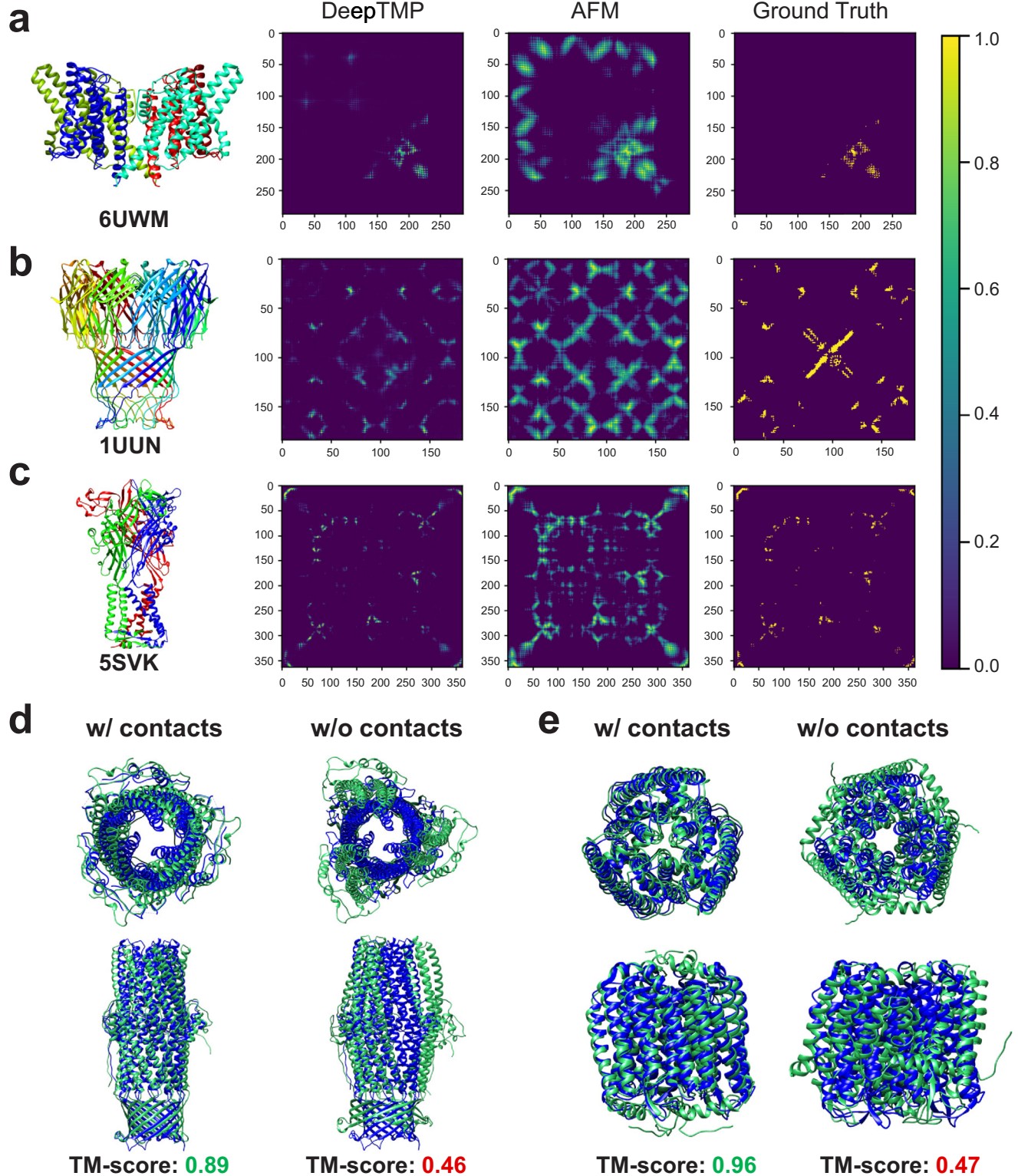

**Fig. 8 | Examples of predicted inter-chain contact maps and contact-assisted docking. a–c** The predicted inter-chain contact maps by DeepTMP, AlphaFold-Multimer (AFM), and ground-truth of native complexes for three example targets, 6UWM (**a**), 1UUN (**b**), and 5SVK (**c**). For reference, the native structures of three targets are also shown on the left. **d, e** The top predicted complex structures of our protein-protein docking program HSYMDOCK with (left panels) and without (right panels) using the predicted inter-chain contacts by DeepTMP on two targets, 4K7R (**d**) and 2JAF (**e**). The upper row is the top view and the lower row is the side view.

where $r_{ij}$ is the distance between two contacting residues. The $E_0$ represents an empirical constraint energy between two residues for a predicted contact and is set to −100.0 kcal/mol[30]. Similar to Dee-pHomo2.0, we only considered the top 10 predicted contacts with a contact probability threshold of >0.65 to minimize the number of false positive contacts. To mimic real experiments, we used the AlphaFold2-predicted monomer structure as input for docking. Figure 8d, e shows the docked complex structures with and without including the

predicted contacts by DeepTMP using our HSYMDOCK program on two targets, 4K7R and 2JAF. It can be seen from the figure that with inclusion of predicted inter-chain contacts, HSYMDOCK is able to construct the correct complex structures with high TM-scores of 0.89 and 0.96 for targets 4K7R and 2JAF, respectively. By contrast, without considering the inter-chain contacts, HSYMDOCK leads to wrongly predicted complex structures with low TM-scores of 0.46 and 0.47 for the two targets, respectively. These results suggest the value of DeepTMP in building complex structures.

## Discussion

In this study, we have developed a transfer learning-based method for predicting the inter-chain contacts of transmembrane protein complexes by transferring the knowledge gained from the initial training model which is trained on a large data set of soluble protein complexes. DeepTMP is extensively evaluated on the TMP test sets of 52 transmembrane protein complexes, and compared with the other five methods including DeepHomo2.0, CDPred, GLINTER, DeepHomo, and DNCON2_Inter. It is shown that DeepTMP achieves the best performance among the compared methods and predicts the inter-chain contacts with the precisions of 82.2% and 68.4% for the top 10 and L predicted contacts, respectively, compared with 48.7% and 31.8% for DeepHomo2.0, 48.5% and 33.8% for CDPred, 38.0% and 27.7% for GLINTER, 27.7% and 15.7% for DeepHomo, and 13.3% and 7.9% for DNCON2_Inter on the test set. These results demonstrate the accuracy and robustness of DeepTMP. The better performance of DeepTMP than the other methods is attributed to not only the use of Resnet-Inception and geometric triangle-aware modules in the network architecture but also the inclusion of transmembrane protein complexes in the training set.

In addition, we also compare DeepTMP with the initial training model (IT_Model) to demonstrate the importance of transfer learning. Through the comparison of the different impacting factors including the proportion of transmembrane regions and the order of symmetry, it is found that DeepTMP can retain the similar physical interaction learned from the pre-trained model such as interfacial hydrophobic interactions, and capture the characteristics of transmembrane protein complexes. Furthermore, we directly train a deep learning-based model without transfer learning on the TMP training set and compare it with DeepTMP to illustrate the importance of the pre-trained model. We also investigate the impact of different features to verify the robustness of DeepTMP. Moreover, we study the performance of DeepTMP on different topologies which have different mechanisms of interactions and structural characteristics, compared with the other methods. It is shown that DeepTMP can effectively predict the correct inter-chain contacts regardless of topologies. Finally, it is revealed that DeepTMP is to some extent able to distinguish between monomers and oligomers as well as between dimers and higher-order complexes according to the maximum predicted contact probability. It is anticipated that DeepTMP will serve as an indispensable tool for the inter-chain contact prediction of homo-oligomeric transmembrane proteins.

## Methods

### The overall framework of DeepTMP

Figure 1 shows the overall framework of DeepTMP, which is composed of four important components: (1) The encoding features of DeepTMP including sequence and structure features. (2) The implementation of ResNet-Inception module. (3) The triangle-aware module including triangular update, triangular self-attention and transition layer. (4) The transfer learning module for transmembrane protein complexes.

### The input features and ground truth

The input of DeepTMP includes sequence and structure features. For the sequence features which are part of the features used by the DeepHomo2.0, it contains the position-specific scoring matrix (PSSM)

feature, two types of direct co-evolution features calculated by CCMpred[51], and the sequence representation and multi-head attention matrix generated by protein language model ESM-MSA-1b. The structure feature only includes the intra-protein distance map calculated from the monomer structure of the homo-oligomeric transmembrane proteins.

The above sequence features are all produced from the multiple sequence alignment (MSA). Specifically, we first generate the corresponding MSA by searching the sequence database. Since we focus on the inter-chain interaction on homo-oligomeric transmembrane proteins composed of identical sequences in this study, we only need to run the HHblits software to search the homologous sequences of the monomer sequence. For the hyperparameter of HHblits[52], we set the maximum pairwise sequence identity to 99% for the Uni-Ref30_2020_03 database[53] and applied an E-value of 0.001. Then, the PSSM matrix was calculated by the LoadHMM script in the RaptorX-Contact package[13], with a dimension of L×20, where L is the length of monomer sequence. Afterwards, the raw scores and average product correction of direct coupling analysis (DCA-DI and DCA-APC) were calculated by CCMpred, with a dimension of L×L×1 for each of them. Finally, we used the hhfilter[54] to select the most diverse sequence with the diff 512 option for the MSA. The filtered 512 sequences in the MSA are fed to the pre-trained ESM-MSA-1b model to generate the ESM-MSA-1b vector embedding and ESM-MSA-1b row attentions, with the dimensions of L×768 and L×L×144, respectively. It is noted that the above features are generated from the MSA of monomer sequence. Since it is trivial for homo-oligomers to generate the paired MSA, we used the monomer MSA to substitute the paired MSA of complexes and generated the same features.

For the structure feature of monomer structures, we applied the radial basis function to convert the intra-protein distance of monomer as input features[55]. The formulation is described as follows:

$$f(d) = e^{-\left(\frac{d-d^k}{\sigma}\right)^2} \tag{4}$$

where $d$ is the intra-protein distance. $d^k$ is a hyperparameter, which ranges from 2 to 22 Å and is divided into 64 bins with a variance $\sigma = 0.3125$ Å.

For the ground truth of inter-chain contact map, we defined a pair of residues from different chains as being contacted if any two heavy atoms of the two residues are within 8 Å. Since the indistinguishability of the chains in the homo-oligomeric transmembrane proteins, we have implemented symmetrical operations during the evaluation of inter-chain contact predictions.

### The implementation of ResNet-inception module

Previous methods use ResNet module to avoid the explosion of gradient and a large kernel size to capture the long-range interaction between two residues. Recently, several methods combine ResNet and Inception to increase the effective receptive field of the network with fewer parameters[56,57]. As such, before the triangle-aware module, we use four ResNet-Inception modules to preliminarily capture the intra-protein and inter-chain interaction.

### The application of triangle-aware module

The state-of-the-art AlphaFold2[58] proposes the evoformer module including the intra-molecular triangular multiplication update and triangular self-attention, which can reduce the unsatisfied region by the geometric triangular inequality. Therefore, we have also applied the triangle-aware module after the ResNet-Inception module to reduce the geometric inconsistency and more effectively capture the correct inter-chain interactions. The triangle-aware module includes a triangle update, two triangle self-attention, and a transition layer (Fig. 1d).

The triangle-aware module requires three kinds of input features including receptor, ligand, and complex features, which are the

corresponding output hidden features generated by the last ResNet-Inception module. The receptor features are the same as ligand features due to the fact that homo-oligomeric TMPs have identical sequences. In addition, the architecture can also be applied to the case of heterodimer. In that case, the receptor, ligand, and complex features are represented as $r_{ij} \in \mathbb{R}^{L1 \times L1 \times d}$, $l_{ij} \in \mathbb{R}^{L2 \times L2 \times d}$ and $z_{ij} \in \mathbb{R}^{L1 \times L2 \times d}$, where $d$ is a hyperparameter with a value of 64. Therefore, we also used an inter-molecular triangle multiplicative mechanism to update the interchain residue-residue interaction by treating one monomer as the receptor and the other as the ligand. Namely, if residue A of the receptor contacts with residue C of the ligand, the residue B away from residue A in the receptor will not contact with residue C due to the distance constraint.

The triangle update utilizes an attention mechanism on the pair representations of three residues to enforce that the pair representation satisfies the geometric triangle inequality. The detailed formulations of the triangle update are as follows.

$$z_{ij}'^{(k)} = \text{Linear}\left(\phi\left(z_{ij}^{(k)}\right)\right); z_{ij}''^{(k)} = \text{Linear}\left(\phi\left(z_{ij}^{(k)}\right)\right) \qquad (5)$$

$$r_{ij}' = \text{Linear}\left(\phi\left(r_{ij}\right)\right); l_{ij}' = \text{Linear}\left(\phi\left(l_{ij}\right)\right) \qquad (6)$$

$$\tilde{z}_{ij}^{(k)} = z_{ij}^{(k)} + \varphi\left(\sum_{m=1}^{L} r_{im}' z_{mj}'^{(k)} + \sum_{n=1}^{L} z_{in}''^{(k)} l_{nj}'\right) \cdot \phi\left(z_{ij}^{(k)}\right) \qquad (7)$$

where $k$ is the number of blocks in the triangle update ($z_{ij}^{(1)} = z_{ij}$), the function of $\phi$ is a sigmoid function followed by a linear transformation, and the function of $\varphi$ is a layernorm function followed by a linear transformation.

Next, the triangle self-attention is applied with an attention mechanism to calculate the relative strength of the interaction between the pairs of residues. First, we used a multi-head attention on the row dimension which considers the pair representations between one residue of the receptor with all residues of the ligand. The corresponding formulation is described as follows,

$$q_{ij}, k_{ij}, v_{ij} = \varphi_q\left(\tilde{z}_{ij}^{(k)}\right), \varphi_k\left(\tilde{z}_{ij}^{(k)}\right), \varphi_v\left(\tilde{z}_{ij}^{(k)}\right) \qquad (8)$$

$$w_{ijm} = \text{softmax}\left(q_{ij}^T k_{im}\right) \cdot g(d) \qquad (9)$$

$$g(d) = \exp\left(-\frac{d^2}{2\lambda^2}\right) \qquad (10)$$

$$\dot{z}_{ij}^{(k)} = \tilde{z}_{ij}^{(k)} + \varphi\left(\sum_{m=1}^{L}(w_{ijm} v_{im}) \cdot \phi\left(\tilde{z}_{ij}^{(k)}\right)\right) \qquad (11)$$

where $q_{ij}$, $k_{ij}$, $v_{ij}$ is the linear transformation of the output from the triangle update, $d$ is the intra-protein distance, and $\lambda$ is set to 8 Å. Second, we also updated the pair representation in the column dimension. In addition, we used the Naccess program[59] to calculate the solvent-accessible surface area of the monomer and defined the surface residues. Then, we used the mask mechanism like transformer for the surface residues in the triangle update and triangle self-attention.

After that, we used the transition layer which includes two layers of linear transformation, to update the pair representation $\dot{z}_{ij}^{(k)}$ as the input of the next triangle-aware block.

## Transfer learning

The transmembrane protein complexes have similar physical interactions with soluble protein complexes in the interface, such as hydrophobic interactions. For example, the helix-helix interaction regions in transmembrane protein complexes are similar to those found in the interface of soluble proteins[45]. However, the transmembrane protein complexes have different structural and evolutionary characteristics from soluble protein complexes due to the membrane environment. Therefore, a deep learning-based model trained on abundant soluble protein complexes, which learns the common physical interactions, will be suitable as the initial training model for transfer learning. In this study, we used a large data set of homo-oligomers with C2-symmetry screened from our previous work to train the initial model. Afterwards, we employed transfer learning to extract the effective information of transmembrane protein complexes for the pre-trained model on a small training set of transmembrane protein complexes (Fig. 1d). Generally, transfer learning methods utilize two kinds of fine-turning approaches. One is that the weights of some specific layers are frozen and the retrained weights are fine-tuned in the training process. The other is the whole weights of the pre-trained model are fine-tuned without a frozen operator. Here, we trained our transfer learning model without any frozen layers, which achieved a better performance[60].

## Data sets

The data sets for initial training were obtained from our previous DeepHomo study[29]. Specifically, DeepHomo collected all the homo-oligomeric proteins with C2-symmetry type in the PDB and clustered them with sequence and structure criteria into a data set with 4132 non-redundant homo-dimeric complex structures. We used the same training and validation sets for the initial training of DeepTMP.

The data sets for transfer learning were obtained by downloading the structures from the Protein Data Bank of Transmembrane Proteins (PDBTM) on January 2021, which contains 5692 transmembrane protein complexes. Since we focus on the homo-oligomers of TMPs, the heterologous proteins in the PDBTM were excluded, resulting in 2226 structures. In addition, it is noted that some structures from the PDBTM only have one chain even though they are annotated with high-order symmetries. Therefore, we also removed those target structures and obtained 2020 complexes. Furthermore, we filtered their biological assembles with the following criteria: (i) The chain number of complexes is equal to the order of symmetry; (ii) Any pair of chains in the assembly share >99% sequence identity. Finally, the remained 1907 complexes were clustered by MMseqs2 with a sequence identity cutoff of 30%[61,62], which resulted in 322 clusters. For each cluster, the complex with the best resolution was chosen as the representative. In addition, we excluded the complexes with a monomer sequence length of >1024 because the ESM-MSA-1b pre-trained model is limited with a maximum protein sequence length of 1024. Thus, we had a final data set of 309 complexes which were randomly divided into 185, 62, and 62 structures for the training, validation, and test set, respectively. Since our initial training data set is filtered by the interface, we also excluded those partial structures from the test set using the following structure criteria: (i) The maximum area of the interface between any pair of chains in the complex is <500 Å$^2$; (ii) The number of contacts in the interface is <10. After being filtered by those criteria, there were 52 transmembrane protein complexes in the final test set (Supplementary Data 10).

Since we used the MMseqs2 to remove the redundant sequence on the homodimer data set and the TMP data set separately, it may have some redundant sequences between the training/valid data set of homodimers and the test set of TMPs. Therefore, we used the MMseqs2 with an E-value of 0.1 to remove the redundant sequences

from the training and valid sets of homodimers for the initial model, which results in 3079 and 271 targets in the two sets, respectively (Supplementary Data 10). In addition, we also used an E-value of 0.1 to remove the redundant sequence between the TMP training/valid sets and the TMP test set. As a result, the final training and valid sets for transfer learning contain 100 and 38 transmembrane homo-oligomers, respectively (Supplementary Data 10).

### Implementation and training

For the initial training model, we used the Focal Loss as the loss function. We trained the model by using the Adam optimizer with pytorch (v1.8.0) and python (v3.7.0) on A100 GPU. For other hyper-parameters, we used a mini-batch size of 1, a learning rate of 0.001, and a dropout rate of 0.1. Since the triangle self-attention layer would occupy a large GPU memory, we set a maximum length of 256 amino acids in the training process due to the GPU memory limitation. Specifically, for the target proteins with a sequence length of >256 residues, we used a window of 256 and a stride of 1 to scan the sequence and collect the fragments which have the maximum inter-chain contacts. After that, we randomly chose one cropped sequence to represent the protein sequence. For the transfer learning stage, we loaded the initial training model and re-trained the network with the transmembrane protein complexes.

### Reporting summary

Further information on research design is available in the Nature Portfolio Reporting Summary linked to this article.

## Data availability

Data that support the findings of this study are available from the corresponding author upon request. A full list with the links of soluble and transmembrane proteins used in this study is available in Supplementary Data 11. The sequence database of Uniref30_2020_03 used in this study is available at https://www.uniprot.org/help/uniref/. The sequence database of Big Fantastic Database (BFD) used in this study is available at [https://bfd.mmseqs.com/]. The source data underlying Figs. 2, 3, 4a, b, 5, 6, 7, Table 1 and Supplementary Tables 1, 2, 3 are provided in the Source Data file. Source data are provided with this paper.

## Code availability

The DeepTMP package is freely available for academic or non-commercial at [http://huanglab.phys.hust.edu.cn/DeepTMP/].

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

## Acknowledgements

This work was supported by the National Natural Science Foundation of China (grants No. 62072199 and 32161133002) and the startup grant of Huazhong University of Science and Technology.

## Author contributions

S.-YH. conceived and supervised the project. P.L. and Y.Y. prepared the data sets. P.L., Y.Y., and S.-Y.H. designed and performed the experiments. P.L., H.T. and S.-Y.H. analyzed the data. P.L., H.T. and S.H. wrote the manuscript. All authors read and approved the final version of the manuscript.

## Competing interests

The authors declare no competing interests.
