## [Peer Review File · Nature Communications]

Deep transfer learning for inter-chain contact predictions of transmembrane protein complexesReviewers' Comments:

Reviewer #1:

Remarks to the Author:

authors have developed a transfer learning-based method for predicting the inter-chain contacts of transmembrane protein complexes by transferring the knowledge gained from the initial training model which is trained on a large data set of soluble protein complexes. Overall, DeepTMP is useful and interesting. The manuscript is written in a clear and understandable manner, but there are some of my concerns. Thus, I recommend the publication of this manuscript after revision.

(1) What are the main contributions of DeepTMP is compared to other methods (CDPred, DeepHomo2.0, and GLINTER)? Is it because the training set contains membrane proteins? Or is it the main contribution of the features or the network architecture?

(2) In the manuscript, DeepTMP was compared with the state-of-the-art methods (CDPred, DeepHomo2.0, and GLINTER), and achieved better performance. If possible, authors might compare DeepTMP with roseTTAFold or AlphaFold-Multimer on some cases.

(3) Authors runs HHblits software to search the homologous sequences of the monomer sequence in UniRef30_2020_03 database. Is there a performance improvement for DeepTMP if a larger library is used, such as Big Fantastic Database (BFD)?

(4) In the last sentence of the first paragraph of conclusion, the author miss a full stop.

(5) If possible, the authors should provide one or two cases using DeepTMP predicted contact to model overall structure of the protein complex to further increase the usefulness of it.

(6) In figure 1a, the input monomer should be more than one.

Reviewer #2:

Remarks to the Author:

Li and Huang developed a new method for predicting inter-protein contacts for membrane proteins. The work is quite interesting. However, there are questions needed to be addressed to make it better.

Major

1) DNCON2_Inter is a recent work that predicts interprotein contacts. Can you make a comparison? [Scientific Reports volume 11, Article number: 12295 (2021)]

2) As DeepTMP is dedicated to transmembrane proteins, what happens to those regions that are extracellular in a transmembrane protein? Would DeepTMP apply to both regions? What is the accuracy when DeepTMP is applied to soluble proteins? How do they compare to those methods trained on soluble proteins?

3) In the training, validation, test sets, what is the distribution of Alpha, Beta, Alpha+Beta? Would a method trained on alpha, can make an equally good prediction for beta proteins, vice versa, as a generalization test?

4) It is a bit surprising that the precision does not always increase with an increase in number of homologous sequences. Can you explain why? Would this differ for alpha, beta, alpha+beta?

5) IT-model is already better than all previous models (Table 1). Can you explain why?

6) For those monomeric transmembrane proteins, would DeepTMP falsely predict them as dimeric proteins with many false positive inter-protein contacts?

7) Can DeepTMP predict its oligomeric state? (dimer, trimer, etc)?

8) Can AlphaFold2 better predict multimeric transmembrane proteins with predicted contacts by DeepTMP?

Minor

9) Maybe it is better to make a comparison to IT/DT-model before comparing to other methods.

We very much appreciate the constructive comments/suggestions from the reviewers. We have conducted necessary computations/analyses and revised our manuscript accordingly. The revised parts in the manuscript are highlighted **in red**. The point-to-point responses to the comments are listed as follows.

Reviewer #1 (Remarks to the Author):

authors have developed a transfer learning-based method for predicting the inter-chain contacts of transmembrane protein complexes by transferring the knowledge gained from the initial training model which is trained on a large data set of soluble protein complexes. Overall, DeepTMP is useful and interesting. The manuscript is written in a clear and understandable manner, but there are some of my concerns. Thus, I recommend the publication of this manuscript after revision.

Response: We thank the reviewer for reviewing our manuscript and giving the valuable comments and suggestions. We have addressed the reviewer's comments and revised our manuscript accordingly.

(1) What are the main contributions of DeepTMP is compared to other methods (CDPred, DeepHomo2.0, and GLINTER)? Is it because the training set contains membrane proteins? Or is it the main contribution of the features or the network architecture?

Response: We thank the reviewer for raising the questions. We have conducted an ablation study to investigate the main contribution to DeepTMP. Namely, we have done a comparison of the baseline model (i. e. DeepTMP) and three ablation models including No_Module (i. e. DeepHomo2_TMP), No_TMP (i. e. IT_Model), and No_Module+TMP (i. e. DeepHomo2.0). It is revealed that the better performance of DeepTMP than the other methods is attributed to the transmembrane protein complexes in the training set more than the Resnet-Inception and geometric triangle-aware features in the network architecture.

[Page 13, last paragraph; Page 14, 1st paragraph]

(2) In the manuscript, DeepTMP was compared with the state-of-the-art methods (CDPred, DeepHomo2.0, and GLINTER), and achieved better performance. If possible, authors might compare DeepTMP with roseTTAFold or AlphaFold-Multimer on some cases.

Response: We thank the reviewer for the valuable suggestion. We have compared the contact maps between DeepTMP and AlphaFold-Multimer on three cases (PDB IDs: 6UWM, 1UUN, and 5SVK). It is shown that DeepTMP not only obtains a higher performance than AlphaFold-Multimer in predicted correct inter-chain contacts on the three cases, but also gives correct predictions for those contacts that are failed by AlphaFold-Multimer.

[Page 19, 1st paragraph; Page 42, Figure 8]

(3) Authors runs HHblits software to search the homologous sequences of the monomer sequence in UniRef30_2020_03 database. Is there a performance improvement for DeepTMP if a larger library is used, such as Big Fantastic Database (BFD)?

Response: We thank the reviewer for the valuable suggestion. We have evaluated DeepTMP using the Big Fantastic Database (BFD) instead of the UniRef_2020_03 database (UniRef). Interestingly, DeepTMP/BFD yields a slightly lower performance than the baseline DeepTMP/UniRef model, although the DeepTMP/UniRef+BFD model recovers the lost performance. The reason for such difference would be attributed to the different preparation method of UniRef and BFD databases, which will result in different MSAs for the same query. It is also revealed that DeepTMP/BFD is quite complementary to DeepTMP/UniRef. Specifically, out of the 52 targets, DeepTMP/BFD performs better for 22 cases, DeepTMP/UniRef performs better for 24 cases, and both models perform comparably for 6 cases.

[Page 13, 2nd paragraph; Page 39, Figure 5c]

(4) In the last sentence of the first paragraph of conclusion, the author miss a full stop.

Response: We thank the reviewer for pointing out the error. It was corrected.

[Page 20, 2nd paragraph]

(5) If possible, the authors should provide one or two cases using DeepTMP predicted contact to model overall structure of the protein complex to further increase the usefulness of it.

Response: We thank the reviewer for the valuable suggestion. We have used our protein-protein docking program HSYMDOCK to model the overall structures of transmembrane protein complexes on two cases (PDB ID: 4K7R and 2JAF) by integrating the predicted inter-chain contacts by DeepTMP. It is shown that with the inclusion of prediction contacts, HSYMDOCK is indeed able to model the correct structures with high TM-scores of 0.89 and 0.96 on the two cases, while HSYMDOCK gives the wrong structures with TM-scores of 0.46 and 0.47 when the predicted contacts

are not considered.

[Page 19, 2nd paragraph; Page 20, 1st paragraph; Page 42, Figure 8]

(6) In figure 1a, the input monomer should be more than one.

Response: Following the reviewer's suggestion, we have modified the number of input monomers in Figure 1a.

[Page 35, Figure 1a]

Reviewer #2 (Remarks to the Author):

Li and Huang developed a new method for predicting inter-protein contacts for membrane proteins. The work is quite interesting. However, there are questions needed to be addressed to make it better.

Response: We thank the reviewer for reviewing our manuscript and giving the valuable comments and suggestions. We have addressed the reviewer's comments and revised our manuscript accordingly.

Major

1) DNCON2_Inter is a recent work that predicts interprotein contacts. Can you make a comparison? [Scientific Reports volume 11, Article number: 12295 (2021)]

Response: We thank the reviewer for the suggestion. We have added the comparison with DNCON2_Inter on the TMP test set with experimentally and AlphaFold2-predicted structures. Since the DNCON2_Inter only successfully predict the 51 targets, we only show the average precision of the 51 targets in Table 1. Overall, DeepTMP achieved a better performance than the DNCON2_Inter.

[Pages 9-11, Section 2.5; Page 34, Table 1]

2) As DeepTMP is dedicated to transmembrane proteins, what happens to those regions that are extracellular in a transmembrane protein? Would DeepTMP apply to both regions? What is the accuracy when DeepTMP is applied to soluble proteins? How do they compare to those methods trained on soluble proteins?

Response: We thank the reviewer for raising the questions. We have added a separate paragraph to analyze the performance of DeepTMP on three different regions, including

extracellular (Extra), transmembrane (TM), and cytoplasmic (Cyto) segments. It is shown DeepTMP achieves the overall best performance with a top L precision of 72.1% for the cytoplasmic region, followed by 66.0% for the transmembrane and 61.7% for the extracellular regions. These results suggest that the inter-chain contacts for both transmembrane and non-transmembrane regions would be evolutionarily conserved in the formation of transmembrane protein complexes and thus can be well predicted by DeepTMP.

[Page 16, last paragraph; Page 17, 1st paragraph; Supplementary Table 3]

In addition, we have also evaluated DeepTMP on the 296 soluble homodimers from the test set of DeepHomo2.0, and compared it with CDPred, GLINTER, DeepHomo2.0, DeepHomo, and DNCON2_Inter with experimentally and AlphaFold2-predicted structures. It is shown that DeepTMP obtains a similar performance with CDPred, and a better performance than the other methods. This indicates that DeepTMP learned the knowledge of transmembrane protein complexes from transfer learning methods and still be robust on soluble proteins.

[Page 11, last paragraph; Supplementary Table 1]

3) In the training, validation, test sets, what is the distribution of Alpha, Beta, Alpha+Beta? Would a method trained on alpha, can make an equally good prediction for beta proteins, vice versa, as a generalization test?

Response: We thank the reviewer for the valuable suggestions. There are 75/12/13, 25/7/3, and 38/3/11 cases for alpha/beta/alpha+beta topologies in the training, validation, and test sets, respectively. Since the scarcity data of Beta and Alpha+Beta topologies in the training set, we only retrained a model with the Alpha-topology transmembrane protein complexes, named as DeepTMP α . Interestingly, DeepTMP α does not perform the best on those alpha targets. DeepTMP α is similar to DeepTMP in the performances of different topologies. Namely, DeepTMP α achieves the best performance on beta targets, followed by alpha+beta and alpha targets. Specifically, DeepTMP α obtains the precisions of 58.2%, 74.5%, and 76.9% for top L predicted contacts on the alpha, beta and alpha+beta targets, respectively.

[Page 15, 2nd paragraph; Supplementary Table 2]

4) It is a bit surprising that the precision does not always increase with an increase in number of homologous sequences. Can you explain why? Would this differ for alpha, beta, alpha+beta?

Response: We thank the reviewer for raising the questions. We have done a careful investigation on this abnormal phenomenon. It is revealed that two alpha targets (PDB IDs: 4JKV and 5GUF) in this MSA range have an extremely low contact density of 0.343 and 0.324, which makes it hard to correctly predict the inter-chain contacts.

Similarly, another alpha target 4OR2 also has a contact density of 0.325, which leads to a relative low average precision for the $\ln(\text{Meff})$ from 7.0 to 7.25. If we excluded these three targets from the test set, the precisions will always increase with the increase of $\ln(\text{Meff})$. Another reason for such abnormal phenomenon is that we used hhfilter to screen the sequences of the original MSA for extracting the ESM-MSA-1b features, which often leads to a drop in the number of effective sequences for ESM-MSA-1b. Therefore, we have instead adopted the number of effective sequences for ESM-MSA-1b to measure the depth of MSA. With this correction, the precision indeed increases with the number of effective homologous sequences (Supplementary Data 4a).

[Page 12, last paragraph; Page 13, 1st paragraph]

For the targets with different topologies, since the above three targets with low contact density belong to Alpha topology, the trend of the precision variation on Alpha topology is consistent with that of the whole test set. The precisions for the other types always increase with the increase of $\ln(\text{Meff})$.

5) IT-model is already better than all previous models (Table 1). Can you explain why?

Response: We thank the reviewer for raising the question. We added a separate paragraph to discuss about this important issue. That is, compared with the other methods, the IT Model implements the Resnet-Inception and geometric triangle-aware modules in the network. On one hand, the Resnet-Inception module can capture long-range interaction by increasing the effective receptive field. On the other hand, the geometric triangle-aware module is also able to consider many-body effects by utilizing an attention mechanism on pair-representations of three residues that satisfy the geometric consistency. As such, the IT Model has a better ability to capture the inter-chain contacts than the other methods.

[Page 14, 2nd paragraph]

6) For those monomeric transmembrane proteins, would DeepTMP falsely predict them as dimeric proteins with many false positive inter-protein contacts?

Response: We thank the reviewer for raising this important question. To address this issue. We first investigated whether DeepTMP can distinguish between monomers and oligomers. It is revealed that compared with the monomers, the oligomers tend to have a higher maximum contact probability. On average, the oligomers have an average maximum contact probability of 0.86, which is significantly higher than 0.64 for the 50 monomers. This suggests that DeepTMP can to some extent distinguish between monomers and oligomers (dimer, trimer, etc.) according to the maximum contact probability. Given this finding, we are able to define a contact probability threshold of 0.77 that can be used as a criterion to distinguish between monomers and oligomers. At this contact probability threshold of 0.77, DeepTMP yields a high precision of 95.8%

for oligomers and a very low false positive rate of 2.01% for monomers within top L predicted contacts, suggesting the potential of DeepTMP in distinguishing between monomers and oligomers.

[Pages 17-18, Section 2.8, Page 41, Figure 7]

7) Can DeepTMP predict its oligomeric state? (dimer, trimer, etc)?

Response: We thank the reviewer for raising the important question. Our investigation shows that among the oligomers, although the complexes with three or more subunits have comparable maximum contact probabilities, they have a higher maximum contact probability than the dimers (Figure 7b). That means that DeepTMP can to some extent distinguish between dimers and the complexes with three or more subunits, but may not distinguish between the oligomeric states of those transmembrane complexes with three or more subunits.

[Page 17-18, Section 2.8; Page 41, Figure 7]

8) Can AlphaFold2 better predict multimeric transmembrane proteins with predicted contacts by DeepTMP?

Response: We thank the reviewer for the valuable comment. Unfortunately, since AlphaFold2 only works with multiple-bin distance-dependent contacts, the one-bin predicted contacts by DeepTMP are not accepted by AlphFold2. Instead, we have used our protein-protein docking program HSYMDOCK to model the overall structures of transmembrane protein complexes on two cases (PDB ID: 4K7R and 2JAF) by integrating the predicted inter-chain contacts by DeepTMP. It is shown that with the inclusion of prediction contacts, HSYMDOCK is indeed able to model the correct structures with high TM-scores of 0.89 and 0.96 on the two cases, while HSYMDOCK gives the wrong structures with TM-scores of 0.46 and 0.47 when the predicted contacts are not considered.

[Page 19, 2nd paragraph; Page20, 1st paragraph; Page 42, Figure 8]

Minor

9) Maybe it is better to make a comparison to IT/DT-model before comparing to other methods.

Response: Following the reviewer's suggestion, make a comparison to IT/DT-model before comparing to other methods.

[Pages 6-11, Sections 2.3, 2.4, and 2.5]

Reviewers' Comments:

Reviewer #2:

Remarks to the Author:

Nice revision. I have no further questions.